# MALDI-MSI as a Complementary Diagnostic Tool in Cytopathology: A Pilot Study for the Characterization of Thyroid Nodules

**DOI:** 10.3390/cancers11091377

**Published:** 2019-09-16

**Authors:** Giulia Capitoli, Isabella Piga, Stefania Galimberti, Davide Leni, Angela Ida Pincelli, Mattia Garancini, Francesca Clerici, Allia Mahajneh, Virginia Brambilla, Andrew Smith, Fulvio Magni, Fabio Pagni

**Affiliations:** 1Center of Biostatistics for Clinical Epidemiology, Department of Medicine and Surgery, University of Milano-Bicocca, 20900 Vedano al Lambro, Italy; g.capitoli@campus.unimib.it (G.C.); stefania.galimberti@unimib.it (S.G.); 2Proteomics and Metabolomics platform, Department of Medicine and Surgery, University of Milano-Bicocca, 20900 Vedano al Lambro, Italy; isabella.piga@unimib.it (I.P.); f.clerici10@campus.unimib.it (F.C.); a.mahajneh1@campus.unimib.it (A.M.); Andrew.smith@unimib.it (A.S.); Fulvio.magni@unimib.it (F.M.); 3Department of radiology, San Gerardo Hospital, 20900 ASST Monza, Italy; daleni@tin.it; 4Department of Endocrinology, San Gerardo Hospital, 20900 ASST Monza, Italy; pincelliangelaida@gmail.com; 5Department of Surgery, San Gerardo Hospital, 20900 ASST Monza, Italy; mattia_garancini@yahoo.it; 6Pathology, Department of Medicine and Surgery, University of Milano-Bicocca, San Gerardo Hospital, 20900 ASST Monza, Italy; virginia.brambilla.vb@gmail.com

**Keywords:** thyroid carcinoma, MALDI-MSI, proteomics, Fine Needle Aspiration

## Abstract

The present study applies for the first time as Matrix-Assisted Laser Desorption/Ionization (MALDI) Mass Spectrometry Imaging (MSI) on real thyroid Fine Needle Aspirations (FNAs) to test its possible complementary role in routine cytology in the diagnosis of thyroid nodules. The primary aim is to evaluate the potential employment of MALDI-MSI in cytopathology, using challenging samples such as needle washes. Firstly, we designed a statistical model based on the analysis of Regions of Interest (ROIs), according to the morphological triage performed by the pathologist. Successively, the capability of the model to predict the classification of the FNAs was validated in a different group of patients on ROI and pixel-by-pixel approach. Results are very promising and highlight the possibility to introduce MALDI-MSI as a complementary tool for the diagnostic characterization of thyroid nodules.

## 1. Introduction

The application of innovative technologies, such as Matrix-Assisted Laser Desorption/Ionization (MALDI) Mass Spectrometry Imaging (MSI), on cytological thyroid specimens is feasible and robust protocols are now available, enabling the molecular signature of different lesions to be characterized [1,2,3]. After the pioneering phase, challenging technical aspects of this approach, such as the interference of hemoglobin and the stability of the samples, were overcome [4,5]. Furthermore, recent technical improvements related to the increased lateral resolution that can be achieved by MALDI-TOF-MS instrumentation enable the detection of small cell subpopulations based on their different protein profiles (i.e., profiles of single cell types), even within regions that are indistinguishable at the microscopic level, highlighting how molecular imaging can be combined with traditional pathology to generate protein signatures and build classification models [6,7,8,9]. Moreover, we have reported that MALDI-MSI is able to distinguish benign and malignant cases in different cytological thyroid specimens [1,2,3]. Moving forwards from the first results obtained using ex vivo cytological smears taken from surgical procedures, the present study applies MALDI-MSI on real Fine Needle Aspirates (FNAs). Even if thyroid FNAs are safe, cost-effective, and efficient diagnostic tools, a significant rate of 20–30% of cases are still indeterminate for malignancy [10]. Ancillary tests like immunohistochemistry and genetics may improve the diagnostic performances but, theoretically, MALDI-MSI could represent an alternative option too [1,2,3]. To the state of the art, MALDI-MSI was restricted to translational research and the reproducibility across multiple centers was the largest remaining obstacle in moving it toward clinical routine. However, promising results came from microbiology, where MALDI-MSI-based classifiers applied the technology in real time in the diagnostic setting. Recently published studies showed the usefulness, advantages, and applicability of MALDI-MSI in different fields of pathology (diagnosis, prognosis, and treatment response) [10]. The preliminary findings of our trial are encouraging especially for the methodological improvement of the protocol and the feasibility of the technique in a particularly complicated field like thyroid cytological specimens.

A statistical model, able to manage the big data that were generated by this high-throughput proteomics approach, was applied for the characterization of thyroid lesions. Our results suggest an association between pathological thyroid features and proteomic information from the FNAs, representing the basis for proteomic signatures that are predictive of disease status.

## 2. Material and Methods

The study was carried out in accordance with the relevant guidelines and regulations; the protocol was approved by the ASST Monza Ethical Board (Associazione Italiana Ricerca sul Cancro Associazione Italiana Ricerca sul Cancro-AIRC-MFAG 2016 Id. 18445, HSG Ethical Board Committee approval October 2016, 27102016). Appropriate informed consent was obtained from all patients included in the study. The present study considers a subset of the consecutive series of subjects who underwent ultrasound (US)-guided FNAs in Monza and were prospectively enrolled in an AIRC-granted clinical study that was powered to discover new markers for the diagnosis of thyroid nodules.

### 2.1. Pathology

US-guided FNAs were performed using a 25-gauge needle at the Department of Radiology, San Gerardo Hospital. One or two passes per nodule were executed and needle washing from every pass was sent for proteomics MALDI-MSI analysis [6]. In blind, pathologists evaluated the corresponding Pap-stained smears for traditional morphological diagnosis and were classified according to the 5-tiered Italian SIAPEC system for reporting thyroid cytopathology [11]. We certified benign Thy2 cases by performing a US examination of patients 12 months after the first US-guided FNA confirming absence of new echographic malignant features, absence of significant increasing of nodule size, absence of nodes metastasis, and no incidence of new suspicious nodules. For malignant cases, histological diagnoses were progressively collected after thyroidectomy to certify the nature of the nodules. The training set included nine subjects with a confirmed benign diagnosis at the pathologist’s morphological examination (hyperplastic nodules/Thy2) and nine patients that were classified as malignant papillary thyroid carcinoma (PTCs/Thy5). An additional 11 patients were involved in the validation set and their cytological classes included: Thy2 (*n* = 4), Thy3 (*n* = 1), Thy4 (*n* = 1), Thy5 (*n* = 4), and 1 PTC metastatic lymph node. Table 1 summarizes the relevant clinical–pathological characteristics for all the cases in the study.

### 2.2. In Situ Proteomics: MALDI-MSI

Needle washing from thyroid FNAs were collected into a CytoLyt solution (20% buffered methanol-based solution, ThinPrep™ 2000 system, CYTYC Corporation, Hologic) and samples were prepared as previously described and finally transferred as a cytospin spot onto ITO glass slides [4,5,12,13,14]. Then, all slides were washed with increased concentration of ethanol (70%, 90%, and 95%) for 30 s each, dried under vacuum for 15 min, and stored at −80 °C until the day of the analysis (mean 24–48 h after the time of biopsy). Before MALDI-MSI analysis, cytological specimens were equilibrated to room temperature, dried under vacuum for 30 min, and the MALDI-matrix sinapinic acid (10 mg/mL in 60:40 acetonitrile:water w/0.2% trifluoroacetic acid) was uniformly deposited, with an optimized method, using the iMatrixSpray (Tardo GmbH, Subingen, Switzerland) automated spraying system. MALDI-TOF-MSI was performed using an ultrafleXtreme MALDI-TOF/TOF (Bruker Daltonik GmbH, Bremen, Germany) in positive-ion linear mode, using 300 laser shots per spot, with a laser focus setting of 3 medium (diameter of 50 μm) and a pixel size of 50 × 50 μm. Protein Calibration Standard I (Bruker Daltonics, Billerica, MA, USA), that contained a mixture of standard proteins within the mass range of 5730 to 16,950 Da, was used for external calibration (mass accuracy ± 30 ppm). Spectra were recorded within the *m/z* 3000–20,000 range. Data acquisition and visualization were performed using the Bruker software packages (flexControl 3.4, flexImaging 5.0). After the analysis, the MALDI matrix was removed with 70% EtOH and the slides were stained with haematoxylin and eosin (H&E), digitally scanned using a ScanScope CS digital scanner (Aperio, Park Center Dr., Vista, CA, USA), and images were coregistered to the MSI datasets in flexImaging for the integration of proteomic and morphological data. Regions of interest (ROIs) containing pathological areas will be comprehensively annotated. Satisfactory specimens should include at least 6 groups of 10 thyrocytes, as per SIAPEC guidelines [15].

### 2.3. Statistical Analysis

Quartiles, ranges, mean, and standard deviation (sd) were calculated for descriptive purposes. The analysis on proteomic data in the training set was performed on ROIs that included only epithelial cells, while for each patient in the validation set, three different approaches were tested: the average spectra generated from the MALDI-MSI analysis, the spectra from each ROI selected by the pathologist, and all the single spectra of the imzML MALDI-MSI analysis (pixel by pixel). The spectra were processed by performing baseline subtraction (median method), smoothing (moving average method, half window width 2.5), normalization (total ion current, TIC), peak alignment, and peak picking (*S*/*N* ≥ 6). The open-source software mMass v.5.5 (http://www.mmass.org) was used to confirm mass spectra alignment. Only peaks with an absolute intensity of less than 0.0003, after TIC normalization, were retained. Intra- and interpatient filters were applied on the detected features in the training set: (i) only the features (*m/z*) detected in at least 25% of the ROIs within the same patient were considered and (ii) the features (*m/z*) that were common to at least 25% of the Thy2 and to 25% of the Thy5 were included in the model and considered to be those most representative of benign and malignant lesions, respectively.

For the two groups in the training set (benign vs. malignant lesions), a logistic regression with a Lasso regularization method was performed [16,17,18]. To select the Lasso penalizing parameter, and to assess the predictive accuracy within the training set, cross-validation was performed. The validation was done in blind from the patient’s histological diagnosis and considering only the features selected by the Lasso model to quantify the probability of malignancy.

Data preprocessing (MALDIquant package) and statistical analyses (glmnet package) were performed using the open-source R software v.3.5.0.

## 3. Results

The cohort of 28 patients included in this study had an average age of 54 years old (sd = 17) and 23 (79%) were females. The average nodule diameter was 20 mm (sd = 9). In the group of patients used in the training phase, the selected ROIs varied in terms of the number of clusters and cells that composed the placards. In the Thy2 cases, an average number of 10 ROIs (range = 5–22, median = 9) was recorded by the pathologist, while, in the Thy5 cases, a mean of 8 ROIs (range = 4–19, median = 6) was selected. To compensate for this variability, equivalent groups of ROIs were generated for each patient: five groups of ROIs for Thy2 cases and four ROIs for Thy5 cases, each comprising from one to seven ROIs. These were then used to calculate the average spectra for the statistical analysis. ROIs from Thy2 lesions had an average number of 9 pixels (range = 3–39, median = 7) while those in the Thy5 had an average of 31 (range = 3–162, median = 13). Therefore, 45 mean spectra were generated for the benign and 36 for the malignant lesions and used for the statistical analysis of the training data. After preprocessing and the two intra- and interpatient filters, 69 features were found to be the most representative of Thy2 and Thy5 lesions; 20 of these were selected from the statistical model as the most discriminant to correctly distinguish samples and quantify their probability of being malignant lesions. Then, the capability of the features included in the model to discriminate benign from malignant lesions was also tested on each single pixel present in the analyzed specimens. This was performed using the same groups of patients included in the training phase. A complete overlap of the cytological diagnosis and MALDI-MSI results was observed. In particular, specimens of the benign group were observed to be very homogeneous (uniformly distributed green color, Figure 1a), indicating that all the protein profiles were similar.

In the validation phase on 11 additional lesions (10 patients), three different approaches were applied based on: (i) average of spectra of the ROIs, (ii) overall average spectra of the entire FNA, irrespective of the morphological selection of the ROIs, and (iii) pixel-by-pixel analysis (Appendix A). The average number of ROIs for the specimens used in the validation phase was 12 (2–25, median 11), with a mean number of pixels for each ROI of 15 (1–208, median 5). The model correctly classified all the benign cases (four Thy2, as shown in Figure 1c, and one morphological Thy3, as shown in Figure 3). In the malignant scenario, three Thy5 cases were particularly challenging due to the paucity of cells (Figure 2e: P_1126) or to a heterogenous background of benign/malignant cells (Figure 2c: P_1084, cytological image not shown) or colloid-rich, cystic variant PTC (Figure 2c: P_1187, cytological image not shown). As a consequence, the proteomic analysis did not identify diagnostic signals of alert at the first screening classifying these samples as benign (Figure 2c). Patients Thy5 P_1149 and Thy4 P_1202, both adequate specimens, were correctly classified based on the distribution of the probabilities to be malignant using both ROIs and pixel-by-pixel data (Figure 2c,h, and Figure 3; Appendix A). Then, an additional experiment was planned to support the hypothesis to justify the incorrect classification using ex vivo specimens. Samples from the same nodules (taken ex vivo after thyroidectomy, as previously described [19]) were now correctly classified as malignant by the model, due to a greater amount of neoplastic clusters that did not limit the analysis (Figure 2d,f,g). Analysis of an in vivo specimen of a metastatic lymph node (P_1188) resulted in a correct classification as malignant based on ROIs but as benign in the pixel-by-pixel classification (P_1188). A specimen collected ex vivo from this lymph node was correctly classified based on both ROIs and the pixel-by-pixel model (Figure 3). Finally, the comparison of the three methodological approaches employed for the validation set highlights improved discriminant power in both the pixel-by-pixel and ROI analyses with respect to when the average spectra of the whole sample was employed (Appendix A). This result underlines the particular strengths of MALDI-MSI that could be exploited to support, as a complementary tool, the fundamental diagnostic role of the pathologist.

## 4. Discussion

### 4.1. Proteomics for the Diagnosis of Thyroid Carcinoma

The development of new diagnostic tools to support cytopathologists in the diagnostic triage of indeterminate for malignancy thyroid nodules can be approached from the alternative perspective offered by proteomics [20,21]. Previous reports enlightened the possibility to apply imaging methods such as MALDI-MSI to cytological specimens to combine the analytical power of traditional morphology and molecular signatures [22]. Preliminary experiments were done using ex vivo specimens taken from surgical samples [19], while in the present study, true needle washing specimens were used. The feasibility of the MALDI-MSI approach to spatially localize proteins in a cancer cell area is enlightened in Appendix A. This represents an intriguing and important methodological step, leading to the recovery of leftover material from the FNAs that can be recovered by washing the needle and stabilizing the cells for two weeks [5]. This procedure allows specimens to be collected from centers that do not have a diagnostic unit with proteomics facilities and then shipped to the referee lab within the following 10 days. In the near future, the more systematic enrollment of patients from multiple centers could ensure the generation of diagnostic libraries containing molecular signatures, which include different malignant and rare histotypes for research purposes.

### 4.2. Big Data and Biostatistics: A Requirement for the Introduction of Proteomics in Clinics

Indeed, the application of proteomics as a routine option for the characterization of challenging cases also requires the development of an enlarged network given that the validation of protocols, biostatistic models, and putative analytical features is related with the interlaboratory reproducibility, standardization of workflows, and diagnostic strengthening of the methods. In particular, with the advent of molecular techniques like next-generation sequencing (NGS) and proteomics approaches (MALDI-MSI), biostatistics models and bioinformatics that can manage big data are necessary for improving the confidence of pathologists [23,24]. Statistical models of cancer at the genomic, proteomic, and transcriptomic levels have proven effective in developing diagnostic and prognostic molecular signatures, as well as in identifying pathogenetic pathways [25]. High-throughput experimental tools allow for the simultaneous measurement of thousands of biomolecules, integrating heterogeneous data into quantitative predictive models to significantly improve cytological diagnoses. Molecular diagnostic workflows can be divided into those that employ unbiased statistical inference and those that also incorporate a priori constraints of specific biological interactions from data [26]. In the present study, a diagnostic model was trained using clear-cut benign or malignant cases to identify specific discriminant features to be tested in the validation phase. Three different approaches were used: the analysis of groups of ROIs that the pathologists selected using morphological criteria, a pixel-by-pixel approach, and examination of the average spectrum of the whole sample.

### 4.3. Training Phase: Features Selection for Benign and Malignant Thyroid FNAs Discrimination

The histograms in Figure 1; Figure 2 show how the probability of being malignant can be effectively represented with curves and the samples from FNAs should not pass the diagnostic proteomic triage whenever a signal of alert was pointed out. After the application of filters, biostatisticians designed a combination of features that was able to correctly distinguish all the training cases, in blind, when they were retested. The highest probability to be malignant of 7% (overall mean of the 3rd quartiles = 2.89%, sd = 2.03%) for the Thy2 and the minimum of 28% (overall mean of the 3rd quartiles = 81.81%, sd = 22.66%) for Thy5 were observed in the training phase (Appendix A).

### 4.4. Validation Phase of the Selected Features and Pixel-by-Pixel Classification of Thyroid FNAs

Results obtained in the pixel-by-pixel validation phase showed that all benign lesions, including the Thy3 (later confirmed as benign after surgical resection), had a 3rd quartile value of the probability of being malign below 7%. The malignant lesions had a 3rd quartile above 28% with the exception of specimens with scarce cellularity or a heterogeneous background. These specimens stressed the model due to particularly challenging nodules that were representative of the diagnostic situation characterizing routine thyroid pathology. Samples with issues in terms of quantitative adequacy, haemorrhagic slides, colloid-rich or very heterogeneous FNAs with interspersed macrophages and lymphocytes are all good examples of challenging specimens. In benign lesions, a minimum amount of cells was sufficient to confirm the nature of the hyperplastic goiter and no *signal of alert* was recorded. In the malignant group, three FNAs from histologically proven PTC (Thy5) were not correctly assigned (Figure 2c) due to the quality of the samples taken from the patient. Two different situations were highlighted: samples with paucity of malignant thyrocytes or with high inflammatory or colloidal background. In fact, when the analyses were repeated with samples taken ex vivo from the thyroid of the same patients after surgical removal, they were easily diagnosed as malignant by our diagnostic tool due to the increased quality of the specimens with a greater amount of neoplastic clusters (Figure 2d). An in vivo specimen of a metastatic lymph node was also misclassified as benign only in the pixel-by-pixel classification (P_1188). A possible explanation for this failure could be due to the low number of thyrocytes present in the sample. As a consequence, the correct classification was obtained when using the ROIs, where the background was less impacted by the quality of the spectra, but this confounded the model in the pixel-by-pixel classification. However, the specimen that was collected ex vivo was correctly classified using either the ROIs or the pixel-by-pixel model (P_1188: Figure 3 and Appendix A). This suggests that, once the pathologist certified the presence of a satisfying quantity of neoplastic cells in the washing material, the model also correctly triaged malignant PTC cells in a sample taken from a metastatic lymph node.

## 5. Conclusions

Notwithstanding the consideration that the diagnostic validity of the model needs to be verified in the large cohort of patients that is currently under enrollment, the present study introduces an original methodological approach to build a proteomic diagnostic tool in thyroid cytopathology by taking advantage of MALDI-MSI technology. The next step will be to systematically test the workflow and to putatively identify the most significant features employed by the classification model. The direct consequences of successful results could be the use of MALDI-MSI proteomics as a complementary approach for the characterization of indeterminate for malignancy thyroid nodules. Despite the technical challenges of this study, the application of proteomics and imaging may help to elucidate key biomolecular events and pathways in oncogenic processes [27,28]. Collectively, this represents an important paradigm for both the fundamental characterization of cancer systems and the discovery of molecular targets for diagnostic application.

## Figures and Tables

**Figure 1 cancers-11-01377-f001:**
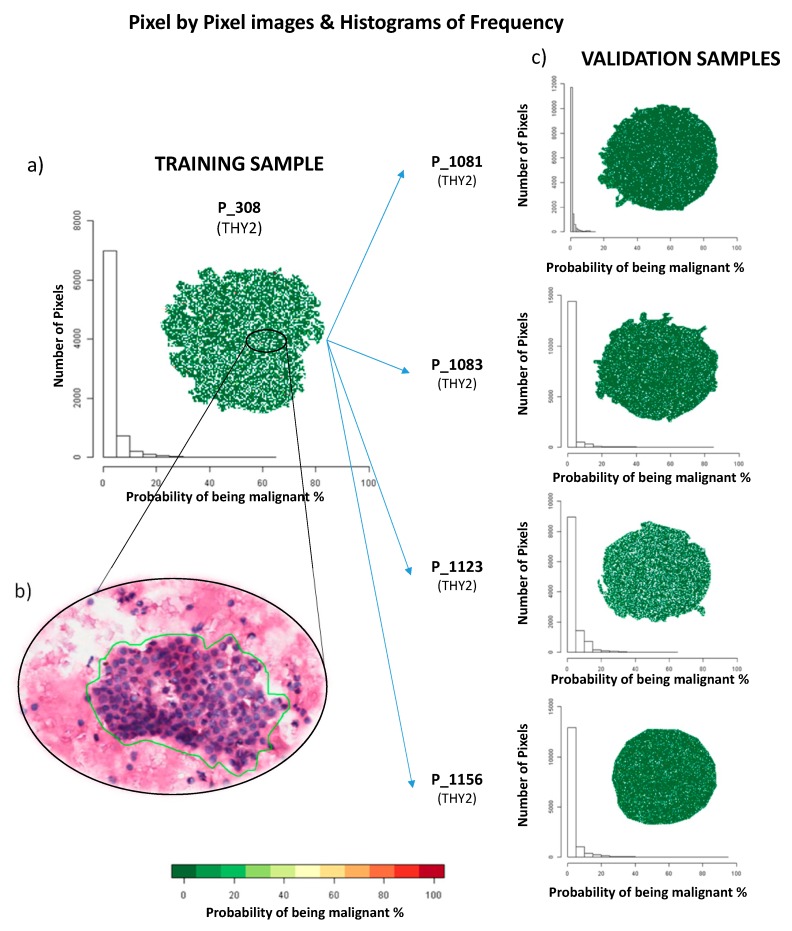
Examples of pixel-by-pixel images and distributions of the probabilities of being malign in the training and validation set of benign Thy2 nodules. (**a**) imzML MALDI-MSI data of the Thy2 P_308 in the training sample; (**b**) haematoxylin and eosin (H&E) staining of P_308; (**c**) validation of Thy2 samples using imzML MALDI-MSI data.

**Figure 2 cancers-11-01377-f002:**
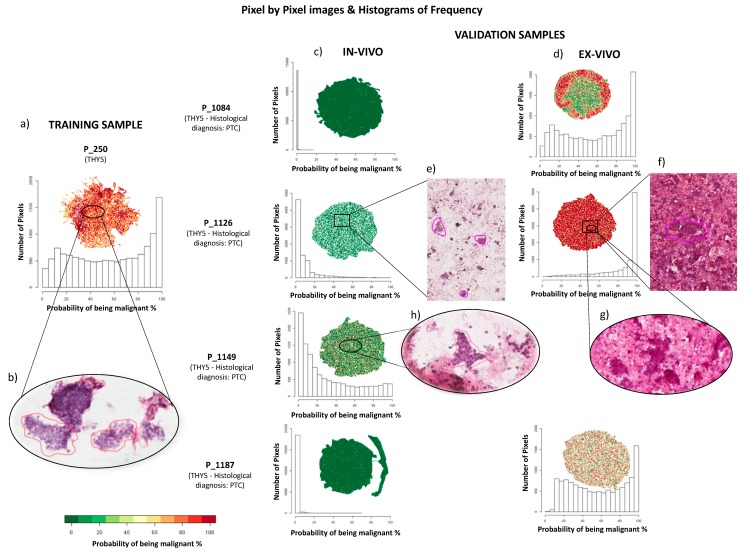
Examples of pixel-by-pixel images and distributions of the probabilities of being malign in the training and validation set of malignant Thy5 nodules. (**a**) imzML MALDI-MSI data of the Thy5 P_250 in the training sample; (**b**) H&E staining image of P_250; validation of (**c**) in vivo Thy5 samples and (**d**) ex vivo Thy5 samples using imzML MALDI-MSI data; (**e**) low cellularity in the H&E staining image of the P_1126 in vivo sample; (**f**) high cellularity in the H&E staining image of P_1126 ex vivo sample and (**g**) a zoom-in of thyrocyte clusters; (**h**) H&E staining image of high-quality cluster of thyrocytes cells of P_1149 in vivo.

**Figure 3 cancers-11-01377-f003:**
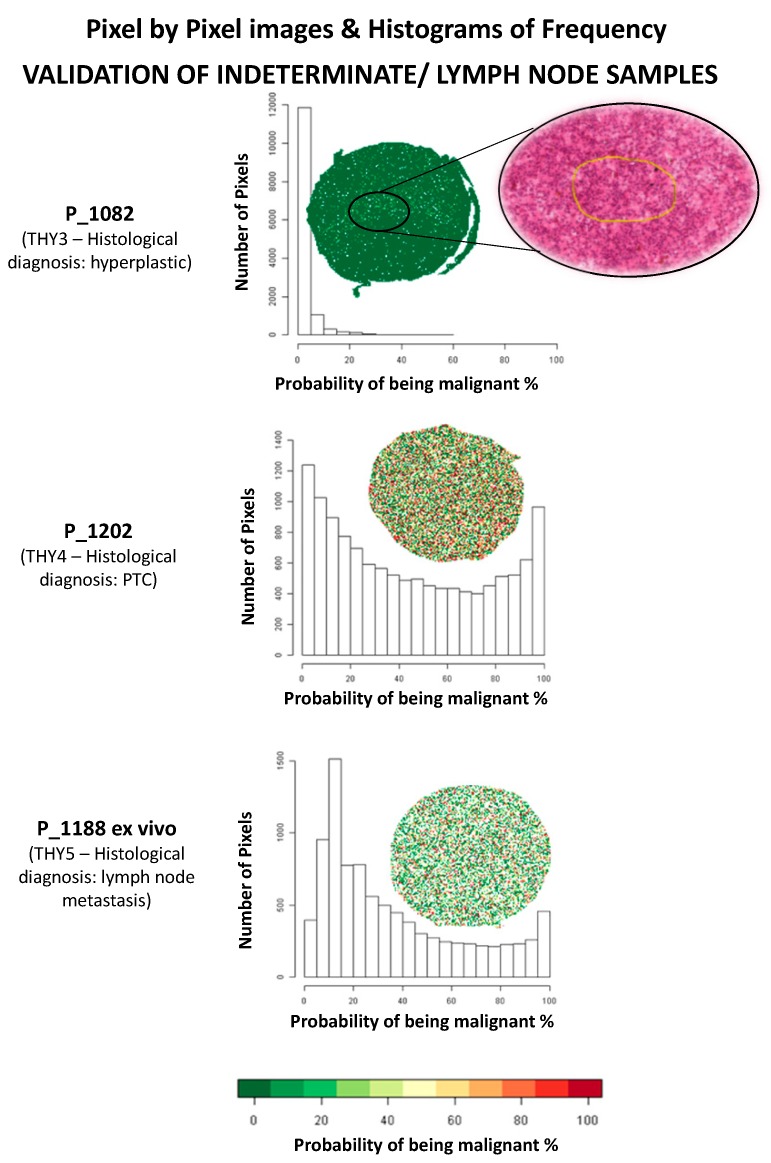
Validation set of indeterminate for malignancy (Thy3), suspicious (Thy4) cases, and metastatic lymph node. Pixel-by-pixel images and distribution of the probabilities of being malignant for each pixel in the MALDI-MSI analysis.

**Table 1 cancers-11-01377-t001:** Clinical information of the lesions and the patients included in the study. Green corresponds to Thy2 hyperplastic nodules; orange corresponds to nodules with an indeterminate for malignancy or suspicious cytological diagnosis; in red, malignant Thy5 cases are listed.

Study Lesion Code	Age (Years)	Sex	Nodule Size (mm)	FNA	Classification at Follow-Up or Histology
***TRAINING SET***					
262	81	F	30	THY2	Hyperplastic
268	81	F	10	THY2	Hyperplastic
302	63	F	15	THY2	Hyperplastic
308	32	F	10	THY2	Hyperplastic
384	71	F	20	THY2	Hyperplastic
475	39	F	25	THY2	Hyperplastic
565	69	M	22	THY2	Hyperplastic
1046	56	F	18	THY2	Hyperplastic
1122	76	F	11	THY2	Hyperplastic
213	48	F	15	THY5	PTC
250	87	F	20	THY5	PTC
436	69	M	14	THY5	PTC
440	45	F	23	THY5	PTC-FV
442	40	F	15	THY5	PTC
992	46	F	13	THY5	PTC-FV
995	61	F	50	THY5	PTC-FV
1012	69	M	18	THY5	PTC-FV
1076	38	F	14	THY5	PTC
***VALIDATION SET***					
1081	79	F	35	THY2	Hyperplastic
1083	49	F	15	THY2	Hyperplastic
1123	36	F	36	THY2	Hyperplastic
1156	53	F	11	THY2	Hyperplastic
1149	30	F	15	THY5	PTC
1084	60	M	11	THY5	PTC-FV
1126	54	M	20	THY5	PTC
1187 *	24	F	25	THY5	PTC
1082	49	F	35	THY3	Hyperplastic
1202	36	M	20	THY4	PTC-FV
1188 *	24	F	25	Metastasis	Lymph node

Legend: M = male, F = female, PTC = Papillary Thyroid Carcinoma, FV = Follicular Variant. * The two lesions are from the same patient.

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
