# Peer review of "MALDI-MSI as a Complementary Diagnostic Tool in Cytopathology: A Pilot Study for the Characterization of Thyroid Nodules"

_cancers, 2019, doi:10.3390/cancers11091377_

Round 1

Reviewer 1 Report

Thanks to the Authors for taking into account my comments. It seems to me that an article in present form can be accepted.

Author Response

Thanks a lot for your evaluation.

Reviewer 2 Report

I recommend publication pending the minor comments below are addressed.

Caption Suppl Fig 1: “spectrum” should read “spectra”

Suppl Figs 1&2: Please add the m/z range values to the mass spectra.

Author Response

Thanks a lot for your suggestions.

We have corrected all the figures as requested.

Prof Pagni

This manuscript is a resubmission of an earlier submission. The following is a list of the peer review reports and author responses from that submission.

Round 1

Reviewer 1 Report

1. The major weakness of the study is the small number of samples taken for analysis.

2. In the Introduction the Authors should describe in greater detail why new methods for diagnosing thyroid nodules are needed. For example, how often it happens that they are incorrectly diagnosed using currently available techniques. It is worth adding information about whether MALDI-MS is routinely used by pathologists for the diagnosis of cancers or other diseases.

3. The description of Table 1 is needed (line 79). The authors should explain the meaning of the colors used in this table.

4. The Discussion should be divided into paragraphs to facilitate reading.

5. References list it is prepared carelessly. It is easiest to avoid such errors using the right software (eg. EndNote).

6. Please explain what publications are hidden after numbers 27 and 28. Now it is unclear.

7. Editorial and language mistakes (examples):

·         The Authors write pixel by pixel (without a dash; eg. line 26) or pixel-by-pixel (with a dash; eg. line 163). Additionally, sometimes this expression is written in italics.

·         Line 61: Extra space in "(US)- guide" should be deleted.

·         Line 62: Extra space in "AIRC- granted" should be deleted

·         Line 83: Full description of the producer of ThinPrep™ system is needed.

·         Line 114: "m/z" should be written in italics.

·         Line 119: "performer [16,17]" should be used instead of "performer[16,17]".

·         Line 129: "(5-22, median 9)" should be used instead of "(5-22; median 9)".

·         Line 149: Extra space in "Thy 2" should be deleted. A similar change should be made in line 179.

·         Line 156: The word "figure" should start with a capital letter.

·         Line 160: "did not" should be used instead of "didn’t". Similar change is needed in line 198.

·         Line 225: The word "table" should start with a capital letter. Similar change is needed in line 247.

Author Response

REV.1

The major weakness of the study is the small number of samples taken for analysis.

We agree with the comment of the referee. However, we explicitly stated in the Title, in Material&Methods (pag. 2) and in the Conclusions (pag. 8) that this is a pilot study reporting the preliminary results on a subset of patients enrolled in a clinical study powered to discover new markers for the diagnosis of thyroid nodules. On the other hand, we included 18 patients in the training set, but they contributed to the analysis with 4-5 ROIs each, for a total of 81 mean spectra. Moreover, the primary aim of this study was to evaluate the potential employment of MALDI-MSI in cytopathology, using challenging samples such as needle washes. A statistical model based on the analysis of Region of Interests (ROIs) was generated and successively, the capability of the model to predict the classification of the FNAs was verified in a different group of patients on both ROI and pixel by pixel approach. Results are very promising and highlight the possibility to introduce MALDI-MSI as a complementary tool for the diagnostic characterization of thyroid nodules. Based on these preliminary results we will continue the enrolment of patients in the AIRC-granted clinical study to reach the target sample size. We hope that our results, even from a small sample of patients, will encourage clinicians to apply MALDI-MSI to other diseases or improve the network of proteomic labs for more impacting multicenter enrollments.

In the Introduction the Authors should describe in greater detail why new methods for diagnosing thyroid nodules are needed. For example, how often it happens that they are incorrectly diagnosed using currently available techniques. It is worth adding information about whether MALDI-MS is routinely used by pathologists for the diagnosis of cancers or other diseases.

The Introduction has been modified according to the referee comments at page 2 . The apppropriate reference was introduced to show MALDI-MSI application in other cancers.

The description of Table 1 is needed (line 79). The authors should explain the meaning of the colors used in this table.

Legend has been modified according to the referee suggestion.

The Discussion should be divided into paragraphs to facilitate reading.

The Discussion has been modified according to the referee suggestion.

References list it is prepared carelessly. It is easiest to avoid such errors using the right software (eg. EndNote).

Done

Please explain what publications are hidden after numbers 27 and 28. Now it is unclear.

Done

Editorial and language mistakes (examples):

The manuscript has been revised.  We fixed all the inconsistencies highlighted by the referee.

The Authors write pixel by pixel (without a dash; eg. line 26) or pixel-by-pixel (with a dash; eg. line 163). Additionally, sometimes this expression is written in italics. Line 61: Extra space in "(US)- guide" should be deleted. Line 62: Extra space in "AIRC- granted" should be deleted Line 83: Full description of the producer of ThinPrep™ system is needed. Line 114: "m/z" should be written in italics. Line 119: "performer [16,17]" should be used instead of "performer[16,17]". Line 129: "(5-22, median 9)" should be used instead of "(5-22; median 9)". Line 149: Extra space in "Thy 2" should be deleted. A similar change should be made in line 179. Line 156: The word "figure" should start with a capital letter. Line 160: "did not" should be used instead of "didn’t". Similar change is needed in line 198. Line 225: The word "table" should start with a capital letter. Similar change is needed in line 247.

Reviewer 2 Report

This manuscript proposes to use proteomics information acquired by mass spectrometry imaging (MSI) from fine needle aspirates (FNAs) to aide in assessing patient diagnostic. Overall, the manuscript is quite convincing even if the cohort analyzed is fairly small. I recommend publication pending the authors consider the comments below.

Readers of Cancers may not be familiar with MSI and I suggest adding some of the raw data to the manuscript (possibly as supplemental material).

1)      To assess the quality of the data, I would have specifically liked to see raw spectral (single pixel) and MSI data from a subset of both benign and cancerous FNAs (see ref 12). Second, for MSI results, I would like to assess the exact correlation between MS signals and the individual cancer cell clumps found after FNA cytospinning and matrix deposition to determine if analyte delocalization has occurred.

2)      I would have also liked to see the list of discriminatory benign / cancer m/z (protein) features allowing prediction of diagnosis. Are these in agreement with previous findings from the authors after investigating thyroid cancer tissue sections by MSI? I would expect most of these to be the same (see again Ref 12).

Supplemental Table 1: It seems that ‘Averages’ were incorrectly calculated. For example, for P_1149 in vivo, the calculated average is 40.22%, (not 16.42%). Please revise.

Author Response

1)      To assess the quality of the data, I would have specifically liked to see raw spectral (single pixel) and MSI data from a subset of both benign and cancerous FNAs (see ref 12). Second, for MSI results, I would like to assess the exact correlation between MS signals and the individual cancer cell clumps found after FNA cytospinning and matrix deposition to determine if analyte delocalization has occurred.

The referee correctly asks for example of mass spectra deriving from single pixel to help the reader to better understand the message of our study. Therefore, we included in the supplementary material mass spectra from 8 single pixel from 4 benign and 4 malignant patients of the training set (SUPPLEMENTARY FIGURE 1). Of course, as expected, we faced with pixel with mass spectra of low quality that were no considered in the pixel by pixel statistical elaboration because they do not meet the criteria described in the section on the Statistical Analysis. Moreover, as the referee knows very well during the MALDI-MSI analysis the mass spectra is obtained using a pixel size of 50x50 mm. However, the pixel sometimes covers a single cell “A” but other times includes part of the cell “A” and part of adjacent cells “B/C” etc. Therefore, the mass spectrum of a single cell several time is not completely related to a specific cell but may depend from the surrounding cells. However, in the pixel by pixel elaboration the classification "malignant Vs benign" is based on the presence of specific feature (m/z and their intensities) independently from their strictly correlation with cells (we should remember that we are not evaluating a tissue with a morphological integrity). Moreover, we optimised our sample preparation and matrix deposition protocols to keep as low as possible the analytic delocalization. Based on our experience with the analysis of fresh frozen and FFPE specimens the delocalization is really very low.

2)      I would have also liked to see the list of discriminatory benign / cancer m/z (protein) features allowing prediction of diagnosis. Are these in agreement with previous findings from the authors after investigating thyroid cancer tissue sections by MSI? I would expect most of these to be the same (see again Ref 12).

The main aim of this study was to evaluate the potential employment of MALDI-MSI in cytopathology, using challenging samples such as needle washes. A statistical model based on the analysis of Region of Interests (ROIs) was generated and successively, the capability of the model to predict the classification of the FNAs was verified in a different group of patients on both ROI and pixel by pixel approach. Results are very promising and highlight the possibility to introduce MALDI-MSI as a complementary tool for the diagnostic characterization of thyroid nodules. Based on these results we will continue the enrolment of patients in the AIRC-granted clinical study to reach the target sample size. We have a broad experience on biomarker discovery and generally a small set of patients are investigated in order to detect signals with discriminant capability before to carry out bigger and more expensive experiments. The results of this interim analysis shows a cluster of discriminant signals (m/z), but when a larger sample of patients will be investigated the cluster could potentially differ from the one we have found (e.g. pilot study PMID: 21136904 and large sample size PMID: 25202906). Therefore, the specific discriminant signals of this pilot study are not important per se, but are relevant in a more general perspective. Moreover, the sample preparation protocols is different from the one used in our previous investigations and, as a consequence, the cluster of discriminant signals are not comparable. At this stage, we did not identify which proteins generate the discriminant signals being this step very demanding. 

Supplemental Table 1: It seems that ‘Averages’ were incorrectly calculated. For example, for P_1149 in vivo, the calculated average is 40.22%, (not 16.42%). Please revise.

“Average” refers to the average spectrum of the patient and not to the mean of the probabilities to being malignant calculated on the ROIs of each patient in the validation set. We have modified Supplemental Table 1 in order to make it clear.

Round 2

Reviewer 2 Report

I am overall very disappointed with the author’s responses to my original review comments:

1) The authors have appropriately shown examples of individual MS spectra (supplemental Figure 1) but have failed to show MSI data from FNA samples. Showing MSI results is crucial to demonstrated analytical feasibility and quality.

2) I do not accept the comments from the authors "Therefore, the specific discriminant signals of this pilot study are not important per se, but are relevant in a more general perspective." I want to assess here that the signals detected are "real" and of significant intensity and consistency to be defined as "statistically significant". Second, my original review comment "Are these in agreement with previous findings from the authors after investigating thyroid cancer tissue sections by MSI? I would expect most of these to be the same (see again Ref 12)." was not innocent. Independently of sample preparation, MS protein markers initially found from MSI analysis of tissue sections are found upon FNA MS analysis often as the top markers! If this is not the case, this is to me very troublesome. I would expect that the markers found in this preliminary study to end up being present and fully validated markers in a follow-up comprehensive study. Note that I did not ask that these (tentative) markers be identified nor validated.

Without these two points addressed, I will not validate this manuscript.

It is also surprising that the original reference 12 has been omitted from the revised manuscript. Clearly, the results from this study (demonstrating the possibility of analyzing FNA's by MALDI MS and comparing these results to tissue sections) bother or contradict findings from the authors.